# Hybrid Mesoporous Silica Nanoparticles Grafted with 2-(tert-butylamino)ethyl Methacrylate-b-poly(ethylene Glycol) Methyl Ether Methacrylate Diblock Brushes as Drug Nanocarrier

**DOI:** 10.3390/molecules25010195

**Published:** 2020-01-03

**Authors:** Abdullah M Alswieleh, Abeer M Beagan, Bayan M Alsheheri, Khalid M Alotaibi, Mansour D Alharthi, Mohammed S Almeataq

**Affiliations:** 1Department of Chemistry, College of Science, King Saud University, Riyadh 11451, Saudi ArabiaBys9314@gmail.com (B.M.A.); khalid.m@ksu.edu.sa (K.M.A.); manalharthi@ksu.edu.sa (M.D.A.); 2King Abdulaziz City for Science and Techenology, Riyadh 11451, Saudi Arabia

**Keywords:** mesoporous silica nanoparticles, polymer brushes, pH responsive polymer, surface-initiated atom transfer radical polymerization

## Abstract

This paper introduces the synthesis of well-defined 2-(tert-butylamino)ethyl methacrylate-b-poly(ethylene glycol) methyl ether methacrylate diblock copolymer, which has been grafted onto mesoporous silica nanoparticles (PTBAEMA-b-PEGMEMA-MSNs) via atom transfer radical polymerization (ATRP). The ATRP initiators were first attached to the MSN surfaces, followed by the ATRP of 2-(tert-butylamino)ethyl methacrylate (PTBAEMA). CuBr2/bipy and ascorbic acid were employed as the catalyst and reducing agent, respectively, to grow a second polymer, poly(ethylene glycol) methyl ether methacrylate (PEGMEMA). The surface structures of these fabricated nanomaterials were then analyzed using Fourier Transform Infrared (FTIR) spectroscopy. The results of Thermogravimetric Analysis (TGA) show that ATRP could provide a high surface grafting density for polymers. Dynamic Light Scattering (DLS) was conducted to investigate the pH-responsive behavior of the diblock copolymer chains on the nanoparticle surface. In addition, multifunctional pH-sensitive PTBAEMA-b-PEGMEMA-MSNs were loaded with doxycycline (Doxy) to study their capacities and long-circulation time.

## 1. Introduction

Mesoporous silica nanoparticles (MSNs) have been studied extensively and applied in various areas, such as colloid chemistry, catalysis, photonics, biosensing, and drug delivery. The great potential of these materials can be attributed to their high rigidity and thermal stability as well as large surface areas, large pore volumes, excellent physicochemical stabilities, and ease of modification [1,2,3,4,5]. MSNs are usually modified on the surface with organic materials, especially polymers, to form silica polymer core/shell nanohybrids [4,5,6,7,8]. Polymer-grafted MSNs combine the advantages of MSNs and organic film to increase the potential applications of these nanomaterials, especially in controlled drug delivery [9,10,11,12,13].

However, controlling the release of a drug from a nanocarrier faces unique challenges, which normally depend on the nanoparticle’s characteristics. Therefore, in order to design a nanosystem with the drug-release kinetics desired for the target applications, it is important to understand the drug-releasing mechanisms [14]. In the past few years, the concept of stimuli-responsive drug delivery systems (i.e., temperature-responsive, light-responsive, enzyme-responsive, or pH-responsive systems) has been developed for tailoring the release profiles [7,15].

Various methods have been used to synthesize silica polymer core/shell hybrid nanoparticles, including surface-initiated reversible addition-fragmentation chain transfer polymerization (RAFT), surface-initiated nitroxide-mediated polymerization (NMP) and surface-initiated atom transfer radical polymerization (SI-ATRP) [16,17,18,19]. SI-ATRP have been used to grow a densely anchored polymer shell with a high degree of control in terms of the size, structure, and uniformity of the polymer chains (polymer brushes) [20,21].

Depending on the chemical composition, a change in the conformation of the polymer chains can be achieved when an external stimuli is applied, such as temperature [22,23,24], solvents [24,25,26], and pH [24,27,28,29]. The synthesis of poly(*N*-isopropyl-acrylamide-cohydroxymethyl acrylamide)-shell–MSNs was reported by Liu et al. [10]. Their results showed that the drug release rate was dependent on the temperature. Liu and co-workers reported the synthesis of hybrid silica nanoparticles grafted onto thermo-responsive poly(ethylene glycol) methyl ether methacrylate (PEGMEMA) which possessed the ability to undergo emulsification–demulsification inversion in response to temperature [30]. An intelligent drug delivery system based on MSNs coated with an ultra-pH-sensitive polymer and poly(ethylene glycol) was synthesized by Chen et al. [31]. The DOX-drug release behavior was reported to be pH dependent with good control. Alswieleh et al. reported the growth of a secondary amine, poly(2-(tert-butylamino)ethyl methacrylate) (PTBAEMA), using SI-ATRP and studied the pH-responsive behavior of these linear brushes [32].

Attention has also been paid to dual stimuli-responsive polymers, which is promising area for smart nanodevices. Further, Chang et al. synthesized pH and thermo dual-responsive poly(*N*-isopropylacrylamide-co-methacrylic acid) core/shell nanohybrids for controlled drug release [33]. Finally, Wu et al. reported the synthesis of hybrid silica nanoparticles with well-defined thermo and pH dual-responsive poly(*N*-isopropylacrylamide)-b-poly(4-vinylpyridine) (SNPs-g-PNIPAM-b-P4VP) via SI-ATRP [34].

To the best of our knowledge, very little work has been done on the synthesis of diblock polymers grafted onto nanoparticles. However, as far as we are aware, no work has been done on fabricating mesoporous silica materials with pH and thermo dual-responsive diblock brushes, as well as a drug nanocarrier. In this study, we have synthesized a PTBAEMA-b-PEGMEMA diblock copolymer grafted onto mesoporous silica nanoparticles (MSNs) via surface-initiated ATRP/ARGET ATRP methods. First, the MSNs were synthesized with amine groups along the inner surface and with pore sizes of ~6.0 nm. Thereafter, PTBAEM was grown on the ATRP initiator-attached mesoporous silica nanoparticle outer surface via SI-ATRP. The PTBAEM end groups can be reinitiated to continue the polymerization on an MSN’s surface with a second monomer, which leads to grafting diblock copolymers onto its surface. The AGET ATRP method allows precise control of the polymer length, thus, it was used to grow a second polymer, namely, PEGMEMA. The pH-responsiveness of the diblock copolymer chains on an MSN’s surface can be studied via Dynamic Light Scattering (DLS).

## 2. Materials and Methods

### 2.1. Materials

An Elga Pure Nanopore 18.2 MΩ system was used to obtain deionized water. Tetraethylorthosilicate (TEOS, 98%), *N*-cetyltrimethylammonium bromide (CTAB, 98%), ammonium hydroxide (28 wt. %), 3-aminopropyltriethoxysilane (APTES, >98%), 2-bromoisobutyryl bromide (BIBB, 98%), triethylamine (TEA, 99%), 2-(tert-butylamino)ethyl methacrylate (TBAEMA, 97%), poly(ethylene glycol) methyl ether methacrylate (PEGMEMA, average Mn 500), copper(I) chloride (>98%), copper(II) bromide (>99%), 2,2′ bipyridine (>99%), methanol (99.8% HPLC grade), ethanol (99.8%, HPLC grade), isopropyl alcohol (analytical grade), toluene (analytical grade), and dichloromethane (DCM, HPLC grade) were purchased from Sigma-Aldrich (Steinheim, Germany). Hydrochloric acid (HCl) was obtained from Fisher Scientific (Loughborough, UK). All the chemicals were used as received. TBAEMA and PEGMEMA were treated with basic alumina to remove the inhibitor and stored at 5 °C before use.

### 2.2. Synthesis of Mesoporous Silica Nanoparticles (without Amine in the Pores)

First, 1.0 g of CTAB was dissolved in 160 mL of deionized water under stirring for 10 min. Concentrated ammonia water (7.0 mL, 28 wt. %) was added to the solution. After that, a mixture solution of n-hexane (20 mL), and TEOS (5 mL) were added to the solution within 20 min under stirring at 35 °C. After stirring for 15 h, the product was collected by centrifugation and washed with deionized water and ethanol.

### 2.3. Synthesis of Mesoporous Silica Nanoparticles (with Amine in the Pores)

Here, 1.0 g of CTAB was dissolved in 160 mL of deionized water under stirring for 10 min. Concentrated ammonia water (7.0 mL, 28 wt. %) was added to the solution. After that, a mixture solution of n-hexane (20 mL), APTES (0.2 mL, 1 mmol), and TEOS (5 mL) were added to the solution within 20 min under stirring at 35 °C. After stirring for 15 h, the product was collected by centrifugation and washed with deionized water and ethanol.

### 2.4. Synthesis of 3-Aminopropyl-Functionalized MSNs (AP-MSNs)

Mesoporous silica nanoparticles (1.5 g) were suspended in a solution of APTES (0.6 mL, 2.5 mmol) in dry toluene (50.0 mL) and heated overnight under reflux. The nanoparticles were then collected by centrifugation, washed twice with toluene and five times with ethanol, and dried under vacuum.

### 2.5. ATRP Initiator Attached on MSNs Outer Surface (BiBB-MSNs)

In a 100 mL flask, AP-MSNs (1.0 g), DCM (40.0 mL), and triethylamine (1.5 mL, 11 mmol) were mixed, then BIBB (1.2 mL, 10 mmol) in 5 mL of DCM was added dropwise to the mixture. The mixture was stirred overnight at room temperature. The modified nanoparticles were centrifuged, washed three times with DCM and five times with ethanol, and dried under vacuum.

### 2.6. Formation of BiBB-MSNs Nanochannels

BiBB-MSNs were treated with a solvent extraction to remove surfactant by adding 1.5 g of BIBB-MSNs to a solution of ammonium nitrate (10 mg/mL) in ethanol (95%) at 80 °C, then having this mixture undergo magnetic stirring overnight. The sample was collected by centrifugation, followed with washing with ethanol three times, and vacuum dried overnight.

### 2.7. PTBAEMA Brushes Grafted on MSNs Surface

BiBB-MSNs (200 mg) were dispersed in a mixture of isopropanol (IPA, 4 mL), water (1 mL), and TBAEMA (2.7 g, 15 mmol) at room temperature, then deoxygenated for 30 min. Cu(II)Br_2_ (2.2 mg, 0.01 mmol) and bipy (30.0 mg, 0.2 mmol) were added to the mixture, which was deoxygenated for 10 min. Copper (I) chloride (CuCl) (2.0 mg, 0.02 mmol) was then added to the mixture. The polymerization reaction was allowed to proceed under a nitrogen atmosphere for 3 h. Then the final product was washed with IPA and ethanol and dried under vacuum.

### 2.8. Copolymer Brushes Grafted on MSNs Surface via AGET-ATRP

PTBAEMA-MSNs (1.50 g) was dispersed in 10 mL of CH2Cl2, CuBr2 (0.1671 g) and bipy (0.702 g) were added, then the mixture was stirred at room temperature. PEGMEMA (1.5 g) and 30 mL of deionized water were added to the mixture at room temperature. The reaction mixture was purged with nitrogen for 30 min to evaporate CH_2_Cl_2_. Ascorbic acid (0.04 mmol) was added dropwise into the deoxygenated solution at 70 °C. PTBAEMA-b-PEGMEMA-MSNs were recovered by centrifugation, washed with ethanol several times, and dried under vacuum.

### 2.9. Doxycycline (Doxy) Loading and pH-Triggered Release

PTBAEMA-b-PEGMEMA-MSNs were dispersed in DI water (1 mg/mL). Next, Doxy (0.4 mg Doxy per mg of fabricated nanoparticles) was added to the suspension, and the pH of the resulting mixture was adjusted to 3.0 with HCl (1M) aqueous solution. The suspension was stirred overnight at room temperature. Thereafter, the pH of the mixture was adjusted to 9.0, and it was stirred for 2 h. Subsequently, the Doxy-loaded PTBAEM-b-POEGMA-MSNs were centrifuged and washed twice with a dilute solution of NaOH (pH = 9.0) to remove the surface adsorbed Doxy. The unloaded Doxy was determined to be in the supernatant based on a standard calibration curve for Doxy.

The encapsulation efficiency (EE) was determined according to the following equation:EE (%) = (Weight of loaded drug/Weight of drug in feed) × 100%,

Doxy-loaded PTBAEMA-b-POEGMA-MSNs were suspended in (pH = 6, pH = 7, pH = 8, and pH = 9) solution at a concentration equal to 0.25 mg NPs/mL. At specific time points, the fabricated nanoparticles were centrifuged and an aliquot of the supernatant (1 mL) was removed and replaced with an equal volume. The amount of Doxy released in the supernatant was calculated based on the calibration curve of Doxy in the same buffer.

### 2.10. Measurement and Characterization

FTIR Spectroscopy: Infrared spectra of all fabricated nanoparticles were obtained in KBr pellets in the 4000–400 cm^−1^ region with a resolution of 4 cm^−1^, using a Thermo Scientific Nicolet iS10. Elemental Analysis (EA): EA was carried out using a Perkin Elmer Series II-2400 analyzer. Scanning Electron Microscopy (SEM): SEM images were obtained using JEOL JSM-6380 LA. The samples were used for the observation without any treatment. Transmission Electron Microscopy (TEM): A drop of dilute sample suspension in ethanol was placed on a copper grid and dried at room temperature. A JEOL JEM-1230 transmission electron microscope was used to collect TEM imaging. Thermogravimetric Analysis (TGA): TGA analyses were carried out on a SII TGA 6300 instrument with 10 °C/min heating rate under N_2_. Dynamic light scattering (DLS) measurements were collected using Zetasizer nano ZS (Malvern Instruments, Malvern, UK) at 25 °C. UV spectra were recorded on Shimadzu (UV-2600) UV–VIS spectrophotometer.

## 3. Results and Discussion

MSNs were synthesized by allowing TEOS to react with a template made of micellar rods (CTAB) in the presence of a pore expander (n-hexane) and APTES in order to functionalize the internal mesopore surface. PTBAEMA-b-PEGMEMA copolymer brushes on MSNs were prepared according to Scheme 1. Due to the hydroxyl groups on the external surface, APTES can be conveniently attached onto the MSNs surface to functionalize the particles with initiator units. The MSNs-Br was obtained via the reaction of the amino groups in MSNs-NH_2_ with 2-bromo-2-methylpropionyl bromide (BIBB) before removing CTAB to avoid reactions in the internal surfaces of the pores. The surfactant was removed using an acidic solution in methanol at 80 °C. SI-ATRP method was used to grow PTBAEMA brushes, followed by the growth of PEGMEMA on MSNs’ external surfaces via AGET-ATRP, to obtain PTBAEMA-b-PEGMEMA-MSNs.

The sizes, shapes, and distribution of the colloidal fabricated nanoparticles were observed using scanning electron microscopy (SEM) and transmission electron microscopy (TEM). SEM images of unmodified MSNs and PTBAEMA-b-PEGMEMA attached MSNs are shown in Figure 1. The image shows that the as-made MSN sample contains nearly nanospherical particles. The as-made MSNs exhibited an average size of 180 nm, as shown in Figure 1A. The SEM image suggest that the particles are coated with a polymer shell which is not visible for the unmodified particles and Zhiping Du made a similar observation (Figure 1B) [30].

TEM images of bare MSNs and PTBAEMA-b-PEGMEMA attached to MSN composites are illustrated in Figure 2. Figure 2A shows that the unmodified nanoparticles are dispersed and have an almost regular morphology with average diameters 170 nm, which is in good agreement with the SEM image (Appendix A). The mesoporous size was estimated to be approximately 6 nm from the TEM image, which is larger than those of typical mesoporous silica nanoparticles due to the effect of n-hexane. In Figure 2B, the observed diameters of PTBAEMA-b-PEGMEMA attached MSNs are larger, and clear core/shell structures due to the polymer chains that are anchored to the MSNs’ external surfaces.

The FTIR spectra were obtained for MSNs-Br, PTBAEMA-MSNs, and PTBAEMA-b-PEGMEMA-MSNs in order to verify that molecules were successfully attached to the surface, see Figure 3. A wide band at 1240–1030 cm^−1^ was observed, which was attributed to Si–O–Si bands stretching the condensed silica network. Further, there was a peak at ~810 cm^−1^ caused by the stretching vibration of Si–O. After linkage of the initiator groups, peaks at ~695 cm^−1^, ~1490 cm^−1^, ~1610 cm^−1^, and ~1565 cm^−1^ were assigned to the APTES functionalization, and a doublet in the vicinity of ~1380 cm^−1^ was due to the isopropyl methyl groups in 2-bromoisobutyrate. The peak at ~2930 cm^−1^ was ascribed to C–H as a –CH_2_– stretching vibration after the attachment of PTBAEMA to the MSN surfaces, and the absorption band at ~1730 cm^−1^ belongs to –C=O stretching. PTBAEMA-b-PEGMEMA-MSNs FTIR spectra are similar to those for MSN-PTAEBMA. However, there is an increase in the peak intensity at ~1730 cm^−1^ as a result of increasing the amount of the (–C=O stretching) functional group present, indicating the successful attachment of PEGMEMA to form diblock polymers.

Elemental analysis was utilized to estimate the number of molecules attached to the surfaces of the MSNs-Br, MSNs-PTEABMA, and PTBAEMA-b-PEGMEMA-MSNs using the measured percentages of carbon, hydrogen, and nitrogen (Table 1). The results clearly show an increase in the percentage of N, C, and H elements on the MSNs-Br (with amine in the inner surface) compared to MSNs-Br (without amine in the inner surface), which confirms the presence of the amino group inside the pores. Furthermore, there was a further increase in the amount of N, C, and H elements when the diblock copolymer was introduced to the surface of the MSNs over the amounts present in the plain MSNs-Br. The ratio of nitrogen content to carbon in PTBAEMA-MSNs was approximately 19%, while it was 14% for PTBAEMA-b-POEGMA-MSNs, further confirming the successful incorporation of the POEGMA into the PTBAEMA-MSNs.

Thermogravimetric analysis (TGA)was utilized to evaluate the successful surface modification of MSNs. TGA was performed under nitrogen atmosphere with heating rate of 10 °C min^−1^ up to 800 °C, as shown in Figure 4. The TGA results indicated that the weight loss was approximately 45% at ~400 °C for as-made MSNs (CTAB and APTES (inner surface)). After CTAB extraction, the weight retention was ca. 8% and ~18% for both MSNs-NH_2_ (inner) and MSNs-Br. The result showed that the APTES content in the internal surfaces of MSNs was ca. 0.13 mmol/g, and the initiator on the external surface was ca. 0.05 mmol/g. TGA revealed that the mass retention of PTBAEMA-MSNs was ca. 40% at ~500 °C. The content of the monomer attached to the outer surface was estimated to be ca. 0.22 mmol/g. The weight loss was ca. 20% at ~550 °C for PTBAEMA-b-PEGMEMA-MSNs samples, which suggests that the amount of PEGMEMA grown on the surface is ca. 0.04 mmol/g. Despite the fact that the estimated polymer layer on the surface of the MSNs was only 15% of the total size of the particles, the polymer weight is considered to be 40% of the total weight of the nanoparticles These results indicate the possibility of a high surface grafting density for the polymers.

The pH-responsive behavior of PTBAEMA-MSNs and PTBAEMA-b-PEGMEMA-MSNs was investigated. The average hydrodynamic diameter (Dh) was measured using dynamic light scattering (DLS) at different pH values ranging from pH 2 to 10, as shown in Figure 5. At pH < 8, the PTBAEM chains are protonated and reach maximum swelling at approximately 1000 nm in thickness (see black dataset in Figure 5). The hydration layer arises on the PTBAEMA-MSNs’ outer surfaces, which results in larger particle diameter measurements. The particles collapsed at a pH of 8 to a diameter of ca. 200 nm due to the deprotonation process. At pH > 8, PTBAEMA-MSNs’ diameters were found to be similar to the diameters estimated by TEM due to hydrophobic PTEABMA chains in this media. The PTBAEMA-b-PEGMEMA-MSNs exhibited pH responsive behavior similar to that of the PTBAEMA-MSNs. It is clear that there was little difference between PTBAEMA-MSNs, and PTBAEMA-b-PEGMEMA-MSNs at pH < 8. When the particles were suspended in a solution with pH > 8, the diblock brushes exhibited a relatively larger particle size of approximately 350 nm (red datasets in Figure 5) due to the neutral hydrophilic PEGMEMA polymer. The changes in the average hydrodynamic diameters of PTBAEMA-MSNs and PTBAEMA-b-PEGMEMA-MSNs at different pH values showed a pKa around 7.7, which is in agreement with the literature [32].

The encapsulation and release of Doxy were investigated as drug delivery vectors for PTBAEMA-b-POEGMA-MSNs (without amine in inner surface) and PTBAEMA-b-POEGMA-MSNs (with amine in inner surface). Doxy was encapsulated in these different MSN samples at an acidic pH of 4. Under these acidic conditions, the PTBAEMA brushes were stretched and in an opened state due to their protonation. Therefore, the Doxy were able to diffuse into the pores of the nanocarriers. The PTBAEMA chains became deprotonated and collapsed on the MSN surfaces upon adjusting the pH to 9.0, forming hydrophobic shells to block the drug inside the channels and allowing Doxy to interact with the inner surfaces of the MSNs (Scheme 2). The encapsulation efficiencies of PTBAEMA-b-POEGMA-MSNs (without amine in inner surface) and PTBAEMA-b-POEGMA-MSNs (with amine in inner surface) were calculated to be 38% and 44%, respectively. The enhanced encapsulation efficiency of PTBAEMA-b-POEGMA-MSNs (with amine in inner surface) compared to PTBAEMA-b-POEGMA-MSNs (without amine in inner surface) indicates that Doxy could be covalently bounded to the interior surfaces of MSNs via amine groups.

Doxy is a tetracycline antibiotic that treats serious bacterial infections, such as cholera, typhus, and syphilis. The pH-triggered Doxy release profile of the PTBAEMA-b-POEGMA-MSNs was investigated in mild acidic (6.0), neutral (7.0), and basic (8.0 and 9.0) pH conditions at room temperature (Figure 6). The unmodified inner surfaces of PTBAEMA-b-POEGMA-grafted MSN samples present clear pH-triggered drug release behavior (Figure 6A). In basic media, an ~20% release was observed after an incubation of 72 h. The drug release was significantly higher at pH 6 and 7 at up to ca. 40%. The acute protonation of PTBAEMA brushes resulted in an open state, facilitating the release of Doxy entrapped by the polymeric shell. The incomplete Doxy release may be ascribed to the physical adsorption of the drug in the inner surfaces of the samples. The release profile for the amino-modified inner surface of PTBAEMA-b-POEGMA-MSNs was investigated in basic media (pH 8 and 9), exhibiting a constant release of ca. 17% after a 72-h incubation period (Figure 6B). Upon decreasing the pH to 6 and 7, the release of Doxy was greater and reached almost 30% after a 72-h incubation period. The observed release behavior is attributed to the attractive and repelling forces between the amino-modified inner surfaces of the PTBAEMA-b-POEGMA-MSNs and Doxy. Overall, the data shows that the release profiles of Doxy loaded in PTBAEMA-b-POEGMA-MSNs with amine in the inner surface followed release kinetics similar to those of PTBAEMA-b-POEGMA-MSNs without amine in the inner surface at both pH 8 and pH 9 with a drug release of ~20%. The lower release level may indicate the blockage of MSN pores by PTBAEMA-b-POEGMA or possibly the strong interaction between the Doxy and the grafted block-polymer, thereby preventing or hindering Doxy release, as shown in Scheme 2. In contrast, the releases of Doxy from PTBAEMA-b-POEGMA-MSNs with amine in the inner surface at both pH 6 and pH 7 was ~40%. Interestingly, this release rate is 20% lower than the release rate for PTBAEMA-b-POEGMA-MSNs without amine in the inner surface. This change may result from the covalent binding between the Doxy molecules and the amino group in the inner surface of the material, as shown in Appendix A.

## 4. Conclusions

In summary, we reported a facile synthesis of well-defined diblock copolymer-grafted on the outer surface of mesoporous silica nanoparticles, i.e., PTBAEMA-b-PEGMEMA-MSNs, via ATRP. Amino-functionalized silane was first attached to the surfaces of MSNs, followed by reaction with BIBB to initiate ATRP, which was then followed by growing PTBAEMA on the nanoparticle surfaces. The retention of the end-group functionality allowed chain extensions from the PTBAEMA-MSNs with PEGMEMA using the AGET ATRP method. The characterizations via TEM, DLS, FT-IR, and TGA confirmed that the PTBAEMA-b-PEGMEMA-MSNs were prepared successfully. DLS was used to investigate the diblock copolymer chain swelling at a pH < 8 which arises from the protonation of the secondary amine groups in the PTBAEMA chains, and the expected size decrease in the nanohybrids occurred when the pH > 8. Doxy was loaded efficiently into the PTBAEMA-b-PEGMEMA-MSN nanostructures, and the drug was released in a relatively controlled pH-triggered manner.

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
