# Peer review of "Hybrid Mesoporous Silica Nanoparticles Grafted with 2-(tert-butylamino)ethyl Methacrylate-b-poly(ethylene Glycol) Methyl Ether Methacrylate Diblock Brushes as Drug Nanocarrier"

_molecules, 2020, doi:10.3390/molecules25010195_

Round 1

Reviewer 1 Report

The manuscript submitted by A.M. Alswieleh and M.S. Almeataq is interesting and can be published in Molecules with minor modifications listed below:

The manuscript and discussion is not clear enough concerning MSN without NH2 in the pores. Scheme 2, MSN (A)

The synthesis should be explained in experimental part, and discussed, few analyses should be given, (Elemental, TGA.....) and compared to MSN (B).

The loading capacity (Mass of loaded drug/mass of loaded drug+mass of MSN) should be calculated.

Reviewer 2 Report

Alswieleh et al. present the formation of mesoporous silica NPs grafted with a pH-responsive copolymer and studied the encapsulation and pH-dependent release of a drug. Although the topic is interesting and nicely visualized, there are many details, inaccuracies and a lack of clarity that require revision of the manuscript.

Abstract: “…(TGA) show that ATRP could provide a higher surface grafting density of polymers.” To what is the higher surface grafting density compared? Furthermore, this point is not mentioned in the Results section. The Introduction is clearly structured. However, the context of drug release appears suddenly without introducing this topic. Later on, why are the examples for dual-stimuli responsive grafted polymers mentioned when the authors don’t discuss dual-stimuli responsive properties of their own work? Mainly, what is new and a remarkable development of the authors work compared to the introduced examples? This should be at least discussed in one of the sections (Introduction, Results or Conclusion). Is it needed to isolate the NP in-between the two polymerization processes? What is the reason that the authors chose two different ATRP-types for the polymerizations of the two monomers? Wasn’t this successful under similar polymerization conditions? Analysis of TEM images (166-173): The big particle in Fig 2A is at least 300 nm in diameter when compared to the scale bar which questions the range 150 nm to 250 nm given by the authors in the text. The images show that there is a large size distribution of the particles. With such a large distribution, it is impossible to derive that the NPs “become larger” with the polymer shell. This would require a statistical analysis of at least 100 and not just 4 particles in the images. It is not obvious for me why the porous structure of the particles is visible for the polymer coated ones (2B), but not for the unmodified ones (2A). For the discussion of the IR spectra, it would be helpful to also show the spectrum of the unmodified particles in Fig. 3. The sample names in Table 1 are confusing (the same for “initiated” in Fig. 4). Are these the same samples like in Figure 3? How did the authors calculate the content (mmol/g-values) from the TGA data? From line 229 on, the authors introduce the particles “without amine in inner surface” and “with amine in inner surface” and this seems to be important for drug loading and release. However, I did not find in the manuscript how these different particles were synthesized. Especially, how is it possible to obtain the particles “with amine in inner surface” without that initiator groups are linked to these during the next functionalization step? Doxy is mentioned from line 229 on. What is Doxy and for what is it used? That authors say “…indicates that Doxy could be covalently bounded to interior surface of MSNs via amine groups” (240). Can the authors compare this with the structure of Doxy to conclude if this interaction is possible or not? The authors do not mention and discuss the significant difference from Fig. 6 A and B in the text (246-258). This should be easy to discuss together with Scheme 2. The sentence “The observed release behavior is attributed to the attractive and repelling forces between amino modified inner surface of PTBAEMA-b-POEGMA-MSNs and Doxy.” is unclear for me, especially “attractive and repelling forces”. “ATRP initiator-functionalized silane” (267) is not correct compared to Scheme 1 and the rest of the article Conclusion: The observed difference from Fig. 6 should be important and relevant to mention, or? Can the authors give an estimation if the pH range used for drug release is interesting and usable for biomedical applications or what other use they have in mind? The wording needs to be revised with respect to scientific sense (or sometimes grammar), for example: “APTES can be conveniently anchored onto MSNs to functionalize the initiator” (150) à to functionalize the particles with initiator units “PTBAEMA brushes on was grown” “In Figure 1B, MSNs-Br is coated with outer layer of PTBAEMA-b-PEGMEMA-MSNs…” (160) à the statement of the sentence should probably be that the SEM image suggest that the particles are coated with a polymer shell which is not visible for the unmodified particles and Zhiping Du made a similar observation “and nothing are seen around the unmodified nanoparticles” 176: TEM instead of SEM “After initiation step, peaks” (180) à Does that mean after linkage of the initiator groups? It is bands and not peaks in IR spectra “when heating under a nitrogen” (197) “MSNs collapsed at ca. 200 nm” (217) à the meaning should be probably that the particles collapsed at a pH of … to a diameter of

Round 2

Reviewer 2 Report

Thanks to the authors to the revision. Some of my questions are adequately addressed now, for some others I want to explain what I meant:

I fully agree that it is interesting to synthesize and study dual stimuli responsive polymers grafted to NPs. However, the problem is the concept of the authors: why do they need dual stimuli responsive NP systems when they only use pH during their characterization? If the authors use dual stimuli responsive polymers as the Motivation in the Introduction, the authors should also demonstrate in the Results that their NPs are indeed dual stimuli responsive, but the characterization of the thermo responsive properties is missing. Could the release also be triggered by a thermoresponsive effect? In addition, it has to be Wu instead of Lei et al. in line 65.    I understand the authors and more images could be shown in a Supporting Information. However, it would be still much more professional if the size and the error range would be estimated by determining the diameter of many particles using for example Image J.

11. From my point of view, a sentence explaining about Doxy should be included in the manuscript.

12. The authors could refer to the Supporting Information and show their Scheme in the SI. This would also have the advantage the readers can look for the structure of Doxy. In the last sentence (line 301), the authors mention electrostatic interaction instead of the above mentioned covalent binding.

14. The sentence in the manuscript is unchanged and I suggest to at least add something like (pH smaller 6) / (pH higher 6) if not the whole explanation.

15. To be honest, several things are still as before.
